# Pretreatment Serum Levels of IL-1 Receptor Antagonist and IL-4 Are Predictors of Overall Survival in Multiple Myeloma Patients Treated with Bortezomib

**DOI:** 10.3390/jcm11010112

**Published:** 2021-12-26

**Authors:** Damian Mikulski, Paweł Robak, Ewelina Perdas, Edyta Węgłowska, Aleksandra Łosiewicz, Izabela Dróżdż, Dariusz Jarych, Małgorzata Misiewicz, Janusz Szemraj, Wojciech Fendler, Tadeusz Robak

**Affiliations:** 1Department of Biostatistics and Translational Medicine, Medical University of Lodz, 93-215 Lodz, Poland; damian.mikulski@stud.umed.lodz.pl (D.M.); ewelina.perdas@umed.lodz.pl (E.P.); aleksandra.losiewicz@stud.umed.lodz.pl (A.Ł.); wojciech.fendler@umed.lodz.pl (W.F.); 2Copernicus Memorial Hospital, 93-510 Lodz, Poland; pawel.robak@umed.lodz.pl (P.R.); malgorzata.misiewicz@umed.lodz.pl (M.M.); 3Department of Experimental Hematology, Medical University of Lodz, 93-510 Lodz, Poland; 4Laboratory of Personalized Medicine, Bionanopark, 93-465 Lodz, Poland; e.weglowska@bionanopark.pl (E.W.); djarych@cbm.pan.pl (D.J.); 5Department of Clinical Genetics, Medical University of Lodz, 92-213 Lodz, Poland; izabela.drozdz@umed.lodz.pl; 6Laboratory of Virology, Institute of Medical Biology, Polish Academy of Sciences, 93-232 Lodz, Poland; 7Department of Hematology, Medical University of Lodz, 93-510 Lodz, Poland; 8Department of Medical Biochemistry, Medical University of Lodz, 92-215 Lodz, Poland; janusz.szemraj@umed.lodz.pl

**Keywords:** bortezomib, IL-13, IL-1Ra, IL-4, multiple myeloma, OS, PFS

## Abstract

Multiple myeloma (MM) is characterized by the malignant proliferation of monoclonal plasma cells in the bone marrow with an elevation in monoclonal paraprotein, renal impairment, hypercalcemia, lytic bony lesions, and anemia. Immune cells and associated cytokines play a significant role in MM growth, progression, and dissemination. While some cytokines and their clinical significance are well described in MM biology, others remain relatively unknown. The present study examines the influence on progression-free survival (PFS) and overall survival (OS) by the serum levels of 27 selected cytokines in 61 newly diagnosed MM patients receiving first-line therapy with bortezomib-based regimens. The measurements were performed using a Bio-Rad Bio-Plex Pro Human Cytokine 27-Plex Assay and a MAGPIX Multiplex Reader, based on the Bio-Plex^®^ 200 System (Bio-Rad). The following levels were determined: IL-1β, IL-1Ra, IL-2, IL-4, IL-5, IL-6, IL-7, IL-8, IL-9, IL-10, IL-12, IL-13, IL-15, IL-17, Eotaxin, FGF, G-CSF, GM-CSF, IFN-γ, IP-10, MCP-1, MIP-1α, MIP-1β, PDGF-BB, RANTES, TNF-α, and VEGF. Most patients received a VCD chemotherapy regimen (bortezomib, cyclophosphamide, and dexamethasone). In the final multivariate model, IL-13 cytokine level (HR 0.1411, 95% CI: 0.0240–0.8291, *p* = 0.0302) and ASCT (HR 0.3722, 95% CI: 0.1826–0.7585, *p* = 0.0065) significantly impacted PFS. Furthermore, ASCT (HR 0.142, 95% CI: 0.046–0.438, *p* = 0.0007), presence of bone disease at diagnosis (HR 3.826, 95% CI: 1.471–9.949, *p* = 0.0059), and two cytokine levels—IL-1Ra (HR 1.017, 95% CI: 1.004–1.030, *p* = 0.0091) and IL-4 (HR 0.161, 95% CI: 0.037–0.698, *p* = 0.0147)—were independent predictors of OS. Three clusters of MM patients were identified with different cytokine profiles. In conclusion, serum pretreatment levels of IL-13 and IL-4 are predictors of better PFS and OS, respectively, whereas IL-1Ra pretreatment levels negatively impact OS in MM patients treated with bortezomib-based chemotherapy. Cytokine signature profile may have a potential influence on the outcome of patients treated with bortezomib.

## 1. Introduction

Multiple myeloma (MM) is a neoplasm characterized by aberrant expansion of monoclonal plasma cells with bone marrow involvement, plasma renal impairment, hypercalcemia, lytic bony lesions, and anemia [1,2]. It is the second-most prevalent blood cancer in the United States and Europe. According to the WHO, 176,404 new MM cases and 117,077 deaths due to MM occurred worldwide in 2020 [3]. Multiple myeloma usually occurs around the age of 60 and is more common in men than women [4,5].

Multiple myeloma is an incurable disease characterized by substantial morbidity and mortality [5]. However, survival of patients with MM has improved significantly over the past 15 years following the introduction of several novel therapeutic agents, including proteasome inhibitors (PI), immunomodulating agents (IMiD), and monoclonal antibodies [5,6]. In addition, high-dose chemotherapy with autologous stem cell transplantation (ASCT) significantly prolongs overall survival (OS) in younger patients. The MM treatment options have never been so broad, and new biomarkers are urgently needed to identify an adequate treatment regimen for a particular patient.

Bortezomib was the first-in-class selective and reversible proteasome inhibitor. Such inhibitors have played a vital role in the treatment of MM. Bortezomib itself is approved for use in both the United States (since 2003) and the EU (since 2004) [7]. The drug is known to have antiproliferative and antitumor activity, and although it has been an invaluable tool in treating MM [8], most patients eventually relapse, and many acquire multiple drug resistance [9].

Immune cells and associated cytokines play a crucial role in the growth, progression, and dissemination of MM. While the roles of some cytokines, such as IL-6, TGF-β, and IL-1β, are well described in the MM biology, those of others remain unclear [10,11,12,13]. Previous reports indicated that bortezomib treatment is associated with significantly reduced serum levels of many cytokines and chemokines, including the interleukins IL-1α/β, IL-3, IL-4, IL-5, IL-6, IL-8, IL-12p40, IL-12p70, and IL-13, as well as IFN-γ, TNF-α, MIP-1α/β, GM-CSF, CXCL1, RANTES, VEGF, and Eotaxin [11,12,13]. Our previous study investigated circulating cytokine, growth factor, and chemokine levels and their association with selected clinical and laboratory disease characteristics [11]. Several cytokines were found to be involved in the pathogenesis of MM, and some were useful in predicting treatment response in bortezomib-treated patients; in particular, higher levels of MIP-1α and lower levels of MIP-1β and IL-9 were associated with better response to treatment, while higher IL-1Ra and IL-8 levels were associated with osteolytic bone symptoms.

The aim of the present study was to assess the impact of pretreatment serum levels of 27 selected cytokines on progression-free survival (PFS) and overall survival (OS) in MM patients treated with bortezomib-based regimens as first-line therapy.

## 2. Materials and Methods

### 2.1. Patients

A total of 61 patients with MM were included in the study. All were treated at the Department of Hematology of Copernicus Memorial Hospital, Lodz (Poland), between February 2016 and September 2019. All received bortezomib-based treatment as first-line therapy. The treatment response and progression were recorded according to the classification given by the International Myeloma Working Group [14,15]. All of the patients underwent whole-body, low-dose CT to asses bone disease. The study was conducted according to good clinical and laboratory practice rules and the principles of the Declaration of Helsinki. All procedures were approved by the Ethical Committee of the Medical University of Lodz (the local ethical committee) no. RNN/103/16/KE.

### 2.2. Cytokine Analysis

The method of cytokine analysis has previously been described in detail [11]. Briefly, any patients with preexisting allergies or infections that may result in changes in the cytokine profile were excluded from the study. Two patients were excluded. The first one (ID 22) was excluded due to rheumatoid arthritis, and the second one (ID 58) was excluded due to atopic dermatitis. Blood samples were collected and left undisturbed at room temperature for 30 min. After this time, they were centrifuged at 2000× *g* for 10 min. The resulting serum samples were stored at −80 ° C before analysis. Serum cytokine levels were assayed using the following equipment: a Bio-Rad Bio-Plex Pro Human Cytokine 27-Plex Assay (Bio-Rad Laboratories, Life Science Group 2000, Hercules, CA, USA) and the Bio-Plex^®^ 200 System, MAGPIX Multiplex Reader (Bio-Rad Laboratories, Life Science Group 2000, Hercules, CA, USA), as described previously [12]. Briefly, the Luminex^®^ xMAP^®^ technology based on immunoassay methods is capable of simultaneously quantifying 27 targets: IL-1β, IL-1 receptor antagonist (IL-1Ra), IL-2, IL-4, IL-5, IL-6, IL-7, IL-8, IL-9, IL-10, IL-12, IL-13, IL-15, IL-17, Eotaxin (CCL11), fibroblast growth factor (FGF), granulocyte colony-stimulating factor (G-CSF), granulocyte-macrophage colony-stimulating factor (GM-CSF), interferon gamma (IFN-γ), IFN-γ-induced protein 10 (IP-10), monocyte chemoattractant protein 1 (MCP-1), macrophage inflammatory protein- (MIP) 1α and MIP-1β, platelet-derived growth factor BB (PDGF-BB), RANTES (Regulated on Activation, Normal T Expressed and Secreted, CCL5), tumor necrosis factor α (TNF-α), and vascular endothelial growth factor (VEGF).

The concentration of each cytokine was extrapolated from the calibration curve (individual for each cytokine), determined independently for each experiment (each plate). All samples were analyzed in duplicate.

### 2.3. Statistical Analysis

Categorical variables were expressed as percentages. These values were analyzed using the chi-squared test. Appropriate corrections were used where needed: the Yates correction for continuity or Fisher’s exact test. The Shapiro–Wilk test was used to confirm where the continuous variables had a normal distribution; as they were normally distributed, they were presented as mean ± standard deviation (SD) or median with interquartile range (IQR) depending on the variable distribution. The cytokine concentrations were compared between identified MM patient clusters using ANOVA with the post hoc Tukey’s test, or Kruskal–Wallis test with the post hoc Dunn’s test. Heatmap and hierarchical agglomerative clustering were used to evaluate clusters of patients with similar plasma cytokine profiles. More precisely, hierarchical clustering was performed using the one minus Pearson’s correlation matrix of included cytokines (as a measure of distance) with the complete linkage method.

Survival analysis was conducted using a Kaplan–Meier estimate with univariate and multivariate Cox’s proportional hazards models, as well as the log-rank test. Optimal cutoff values were determined using Cutoff Finder [16]. In Cox’s models, cytokine concentrations were divided by 10 to interpret coefficient values more straightforward. Analyses were conducted using Statistica 13.1 (TIBCO, Palo Alto, CA, USA). *p*-values < 0.05 were considered statistically significant.

## 3. Results

### 3.1. Patients’ Characteristics

The group of patients comprised 32 men and 29 women with a mean age of 61.9 ± 11.3 years (range: 38.3–83.7). A more detailed analysis of their demographic, clinical, and laboratory characteristics is given in Table 1. The distributions according to the ISS stage I-III were 27.9%, 21.3%, and 47.5%, respectively (missing data—3.3%). The predominant isotype of monoclonal protein was IgG kappa (52.5%). Over 65.6% of patients displayed increased levels of beta2-microglobuline (>3mg/L). Regarding CRAB symptoms, most patients (59%) presented with bone disease, 34.4% displayed Hb < 10 g/dL, hypercalcemia was noted in 19.7%, and renal failure was observed in 16.4%. Seven (11.5%) patients demonstrated increased levels of lactate dehydrogenase (LDH) at diagnosis. Cytogenetics data were available for 33 patients (54.1%); the most common abnormalities were amp(1q), followed by del(13q), with frequencies of 18 (54.5%) and 8 (24.5%), respectively. The majority of patients (80.3%) received a bortezomib, cyclophosphamide, and dexamethasone (VCD) chemotherapy regimen. The objective response rate with primary induction treatment for all patients was 83.6%, including 60.6% with ≥ very good partial response (VGPR). Complete response (CR) was achieved by 24 (39.3%) of patients. More than one-third (37.7%) of patients underwent ASCT.

### 3.2. Prognostic Impact of Clinical Variables

Data on PFS and on OS were available for 61 patients. The median PFS and overall survival OS of the cohort were 13.1 and 51.8 months, respectively. Univariate Cox regression analyses of clinical variables for PFS and OS are summarized in Table 2. Only autologous stem cell transplantation (ASCT) during the treatment schedule was clinical factor influencing both PFS (HR 0.33, 95% CI: 0.17–0.64, *p* = 0.001) and OS (HR 0.23, 95% CI: 0.09–0.62, *p* = 0.004). The corresponding Kaplan–Meier plots are shown in Figure 1.

### 3.3. Prognostic Impact of Cytokine Levels

Univariate Cox regression analyses of pretreatment serum cytokine levels for PFS and OS are provided in Table 3. Only IL-13 serum level significantly impacted PFS (HR 0.1398, 95% CI: 0.0272–0.7189, *p* = 0.0185). Regarding OS, five cytokines were significant in univariate analyses: IL-13 (HR 0.0121, 95% CI: 0.0004–0.3271, *p* = 0.0087), IL-1ra (HR 1.0527, CI:1.0177–1.0889, *p* = 0.0029), IL-4 (HR 0.1899, CI: 0.0401–0.9000, *p* = 0.0364), IL-7 (HR 0.6313, CI:0.3988–0.9993, *p* = 0.0497), and PDGF-BB (HR 0.9963, CI: 0.9930–0.9997 *p* = 0.0316). The Kaplan–Meier plots for dichotomized cytokine levels are provided in Figure 2.

The cytokines found to be significant in univariate analysis were entered into a multivariate model building with a stepwise backward Akaike information criterion (AIC) elimination procedure, together with well-established prognostic factors (ISS III, ASCT, age > 70, and presence of bone disease). Briefly, an optimized model is identified by creating multiple models, initially including all selected variables. The variable with the highest *p*-value is eliminated at each step, and an AIC value is calculated for the new model; this process is repeated until no variables remain. The final multivariate models for PFS and OS with the lowest overall AICs are given in Table 4. The analysis confirmed that cytokine levels retained their importance in the simultaneous context of the clinical prognostic factors. The final model for PFS consisted of two variables: IL-13 cytokine level (HR 0.141, 95% CI: 0.024–0.829, *p* = 0.0302), and ASCT (HR 0.3722, 95% CI: 0.1826–0.7585, *p* = 0.0065). The final model for OS included ASCT (HR 0.142, 95% CI: 0.046–0.438, *p* = 0.0007), presence of bone disease at diagnosis (HR 3.826, 95% CI: 1.471–9.949, *p* = 0.0059), and two cytokine levels—IL-1Ra (HR 1.017, 95% CI: 1.0004–1.030, *p* = 0.0091) and IL-4 (HR 0.161, 95% CI: 0.037–0.698, *p* = 0.0147).

### 3.4. Cluster Analysis

A two-way hierarchical cluster analysis was performed to classify cytokine patterns among multiple myeloma patients (Figure 3A). Six cytokines were excluded from the analysis due to high missing values (>15%): GM-CSF (*n* = 41 values below limit of detection), IL-12 (p70) (*n* = 39), IL-10 (*n* = 27), IL-15 (*n* = 25), IFN-γ (*n* = 18), and VEGF (*n* = 10). One patient (ID 50) was excluded, as 11 cytokines in his sample were below the limit of detection.

Two main clusters of cytokines were identified and three clusters of patients. The first cytokine clusters consisted of IL-1ra, IP-10, IL-6, IL-8, G-CSF, and MIP-1α. The second consisted of IL-1β, IL-13, IL-7, IL-4, Eotaxin, IL-2, IL-17, FGF basic, IL-5, TNF-α, MCP-1 (MCAF), IL-9, MIP-1β, RANTES, and PDGF-BB. The detailed level of each cytokine in particular clusters is provided in Appendix A. Generally, the third cluster presented the most distinctive cytokine profile, with decreased concentrations of proinflammatory cytokines (IL-2, IL-7, IL-9, IL-17, FGF basic, PDGF-BB) and chemokines (RANTES, MIP-1β, and Eotaxin). This cluster also demonstrated the lowest mean concentration of regulatory cytokine IL-4 and the highest mean level of IP-10. The first and second clusters had substantially similar cytokine patterns; however, the patients within the second cluster presented with the lowest concentrations of IL-9, MIP-1β, and RANTES, and the highest levels of FGF basic and MIP-1α. The patients in the first cluster were more frequently ISS III stage (*p* = 0.0287). Comparison of clinical variables between clusters is detailed in Appendix A. Patients in the third (“adverse risk”) cluster had significantly shorter OS (HR 2.988 95% CI: 1.1454–7.7444, *p* = 0.0252), whereas patients within the first cluster had the longest median survival (38.7 months vs. 25.0 in the second cluster and 22.3 in the third cluster) (Figure 3B,C). 

## 4. Discussion

Previous studies have shown that different cytokines take part in the pathogenesis, progression, and prognosis of MM [12,17]. The present study evaluated the prognostic significance of 27 cytokine serum levels in 61 previously untreated MM patients receiving bortezomib-based regimens as first-line treatment. Furthermore, three clusters of patients were identified with different overall survival based on cluster analysis.

We observed that a higher serum pretreatment level of IL-13 is an independent predictor of longer PFS in MM patients treated with bortezomib-based chemotherapy. IL-13 is an anti-inflammatory Th2-type cytokine; it has been found to suppress the cytotoxic activities of macrophages and to inhibit pro-inflammatory cytokine production. It is also believed to play a significant role in a number of inflammatory conditions [18]. Moreover, IL-13 plays a role in human osteoclast formation in a lymphocyte-dependent manner [19]. On the other hand, Di Lulo et al. [20] strongly suggest a role for IL-13 in MM progression through upregulation of adhesion molecules and IL-6 secretion by bone marrow mesenchymal stromal cells, which promotes MM cell growth. Previous studies have found that IL-13 upregulates VCAM-1 expression on endothelial cells and increases adhesion molecule expression and IL-6 secretion in fibroblasts [21,22,23]. These observations suggest that bone marrow-derived mesenchymal stem cells (BM-MSCs) can be targets for IL-13 in MM [20]. IL-13 increases adhesive molecule expression and IL-6 secretion by BM-MSCs; these cells demonstrate a similar response to IL-13 stimulation as other cells, including stromal fibroblasts [22]. We hypothesize that IL-13/IL-6 cascade probably is crucial in the process of conversion from MGUS to MM, whereas in active disease, other mechanisms, including the acquisition of harmful genetic changes, start taking over control. This finding is not clear and requires further investigation. It has been documented previously that IL-13 is involved in the stimulation of macrophages for antitumor activity. Following IL13 activation, macrophages have been found to demonstrate large amounts of macrophage C-type lectin receptors (CLRs) [24,25]. A previous preclinical study in mice found IL13 to inhibit the development of T-cell lymphoma and ovarian adenocarcinoma; this appeared to be facilitated by converting tumor-supporting macrophages to cytotoxic effectors [25]. The prognostic value of serum IL-13 level has been evaluated previously in other hematological malignancies. Özyörük et al. [26] report higher serum IL-13 levels in children with lymphoma diagnosed with Hodgkin lymphoma or Burkitt’s lymphoma; however, unlike the present study, they did not find this cytokine to have any prognostic significance.

Among the 27 tested cytokines, IL-1Ra, IL-4, IL-7, IL-13, and PDGF-BB were found to be predictors of OS in MM patients treated with bortezomib-based regimens in univariate models, and IL-1Ra and IL-4 maintained this significance in the multivariate model. However, among the MM patients treated with bortezomib, only IL-4 predicted longer OS in both the univariate and multivariate models: patients with higher level of IL-4 had longer OS (Figure 2). In normal conditions, IL-4 induces TH2 cell, B-cell, mast cell, and eosinophil proliferation, as well as isotype switching for IgE production [27]. Kyrstsonis et al. found that while IL-4 levels were low (median 4 pg/mL) at diagnosis in 75% of MM patients and then rose in remission (median 25 pg/mL), IL-4 values remained stable during the course of the disease in chemotherapy-resistant patients [28]. In addition, Herrmann et al. observed reduced plasma cell growth in MM patients treated with IL-4, probably by inhibition of endogenous IL-6 synthesis [29].

In our study, a high level of IL-1Ra correlated with shorter OS (Figure 2). IL-1Ra is an anti-inflammatory acute-phase protein that competitively inhibits IL-1 activity and specifically inhibits paracrine IL-6 production [30,31]. Previous research has found higher levels of IL-1RA to be associated with bone involvement [12], and MGUS/SM/MM patients have demonstrated significant increases in serum IL-1Ra levels compared to healthy controls [32]. In addition, MM patients have been found to demonstrate elevated IL-1Ra levels in the bone marrow (BM) environment [33] and that IL-1Ra is produced by MM cells. Low post-transplantation IL-1Ra levels have also been found to correlate with engraftment syndrome in patients with plasma cell dyscrasias such as POEMS (polyradiculoneuropathy, organomegaly, endocrinopathy, M-spike, skin changes) [34]. IL-1Ra is a specific blocker of IL-1, which is a crucial factor in the induction of IL-17-producing T-cells in vivo [35].

A phase II trial of patients with smoldering or indolent MM showed improvement in PFS and OS duration after targeted treatment with IL-1Ra (Anakinra), with or without dexamethasone [36]. It was found that IL-1Ra bound to the myeloma proliferative cells and decreased the level of C-reactive protein (CRP), a surrogate for IL-6 production. Seven patients treated with IL-RA alone demonstrated a decrease in the plasma cell labeling index (PCLI), and three patients achieved a minor response (MR) to IL-1Ra alone. When dexamethasone was added, an additional nine patients achieved a PR/MR. This study suggested that IL-1Ra, as a specific inhibitor of IL-1, induced paracrine IL-6 production and was effective in destroying the proliferative myeloma component.

IL-7 and platelet-derived growth factor-BB (PDGF-BB) were also found to be significant predictors of OS in univariate analyses. IL-7 is a cytokine secreted by bone marrow stromal cells. It was previously observed that IL-7 prevents osteoblast formation by decreasing the activity of Runx2/Cbfa1, which is a transcription factor required for osteoblast differentiation [29]. Nierste et al. confirmed the presence of elevated levels of Dickkopf-1 (Dkk-1) and IL-7 in MM patients, and that these were responsible for the osteoblast differentiation from immortalized bone marrow mesenchymal stem cells (MSCs) [37]. In addition, they found that inhibition of Dickkopf-1 (Dkk-1) and IL-7 from MM plasma restored proper osteoblast differentiation in the MSC line. The IL-7 levels did not return to baseline levels in MM patients who are in remission [10].

In the present study, higher platelet-derived growth factor (PDGF)-BB serum level was associated with longer OS in bortezomib-treated patients in univariate analysis. PDGF is an angiogenic factor that can be formed by two A subunits (PDGF-AA), two B subunits (PDGF-BB), or one A and one B (PDGF-AB). PDGF influences c-myc gene expression through the c-myc promoter in a Src-dependent manner [38]. In an in vitro study, PDGF-BB was found to upregulate Myc expression and reduce the melphalan sensitivity of tumor cell clones. Moreover, downregulation of c-Myc protein induced the expression of PDGF-beta receptor molecules and decreased PDGF-BB release. Similarly, an in vivo study found melphalan-resistant MM patients to present overexpressed c-Myc protein and higher serum PDGF-BB receptor levels compared to minor responding patients.

The relationship between pretreatment cytokine serum levels and OS in newly diagnosed MM patients has also been investigated in previous studies. Cheng et al. [39] explored macrophage inflammatory protein 1 alpha (MIP-1α); migration inhibitory factor (MIF); tumor necrosis factor-α (TNF-α); vascular endothelial growth factor-α (VEGF-α); monocyte chemoattractant protein-1 (MCP-1); and soluble interleukins IL-17A, IL-6, IL-21, and IL-10 before treatment. The authors were able to develop a prognostic nomogram using three variables, namely, lactate dehydrogenase (LDH), MIP-1α, and creatinine levels, which accurately predicted the 1 year, 2 year, and 3 year OS of MM patients.

In a retrospective study, serum IL-6 level > 3 pg/mL, serum IL-17A level > 4 pg/mL, and treatment regimens were found to be independent prognostic factors for PFS and OS according to multivariate analyses of selected serum cytokine levels in patients with newly diagnosed MM. The studies cytokines were IL-2, IL-4, IL-6, IL-10, and IL-17A; TNF-α; and IFN–γ [17]. However, IL-4 serum level had no prognostic value, and IL-13 and IL-1RA were not included in the panel of evaluated cytokines in this analysis [17]. In another study, high serum IL-10 was found to predict poor prognosis [40]: the low-IL-10 group (≤169.96 pg mL-1) was found to have an OR rate of 79.2%, and the high IL-10 group (>169.96 pg mL-1), 53.3% (*p* < 0.001). In addition, the patients in the low-IL-10 group had significantly better PFS (3-year PFS rate: 69.3% vs. 13.3%, *p* < 0.001) and OS (3-year OS rate: 93.6% vs. 51.9%, *p* < 0.001) than the high-IL-10 group.

The present study also analyzed the influence of clinical variables on PFS and OS in Cox regression models. Of the tested variables, only the use of autologous stem cell transplantation (ASCT) during the treatment schedule was found to influence both PFS and OS. These observations are in agreement with previous reports [41,42]. High-dose therapy with melphalan followed by ASCT prolongs PFS, even in the era of novel agents [43]. A meta-analysis incorporating large phase 3 trials from January 2000 to April 2017 found HDT/ASCT to be associated with superior PFS than standard-dose therapy (SDT). However, the effect of HDT/ASCT on OS remains ambiguous and was not observed in this meta-analysis [44]. In addition, bone disease at diagnosis also significantly impacted PFS, and bone lesions have been found to have a negative prognostic influence in several previous reports [44,45]. Elsewhere, the presence of extramedullary disease, high FDG uptake, and more than three focal lesions were associated with shorter OS and PFS in a recent meta-analysis [46].

Our study has several limitations. It is not certain that the levels of IL-1Ra and IL-4 are genuinely predictive of response to bortezomib and not simply prognostic (i.e., indicative of a more refractory phenotype). To verify this, the response to the following lines of therapy should be evaluated, especially including other novel drugs. To do so, a much larger cohort of patients is needed with optimal stratification to available treatment options. Furthermore, equating baseline cytokine levels in MM patients with OS is a demanding task, and the results should be interpreted with caution. The cytokine levels should be further evaluated consecutively to identify changes related to the number of subsequent lines of therapy or acquiring resistance to particular drug classes. Even if they are only prognostic, they may help create more accurate biomarkers in the era of novel therapies (including immunotherapy) than our classical tools (e.g., R-ISS).

## 5. Conclusions

Only IL-13 pretreatment serum level was found to significantly impact PFS in newly diagnosed MM patients treated with bortezomib-based regimens. In addition, serum levels of five cytokines—IL-1Ra, IL-4, IL-7, IL-13, and PDGF-BB—influenced OS in univariate analyses. However, only IL-1Ra and IL-4 were found to have independent prognostic value in multivariate analyses. Three clusters of MM patients were identified, with different cytokine profiles and different OS. Our findings indicate that cytokine signature may have a potential influence on the outcome of MM patients treated with bortezomib. However, the clinical and biological importance of these findings require further investigation.

## Figures and Tables

**Figure 1 jcm-11-00112-f001:**
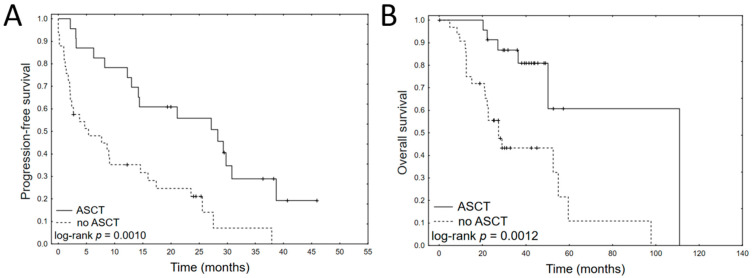
Kaplan–Meier plots of progression-free survival (**A**) and overall survival (**B**) for ASCT in MM patients treated with bortezomib.

**Figure 2 jcm-11-00112-f002:**
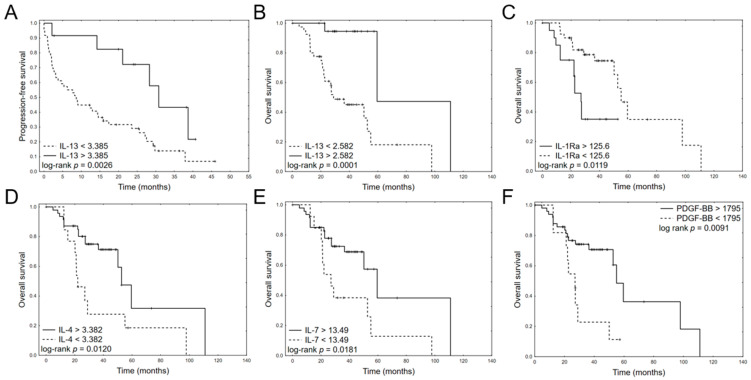
Kaplan–Meier plots for dichotomized significant cytokines in univariate analysis: PFS: IL-13 (**A**) and OS: IL-13 (**B**), OS: IL-1Ra (**C**), OS:IL-4 (**D**), OS: IL-7 (**E**), OS: PDGF-BB (**F**).

**Figure 3 jcm-11-00112-f003:**
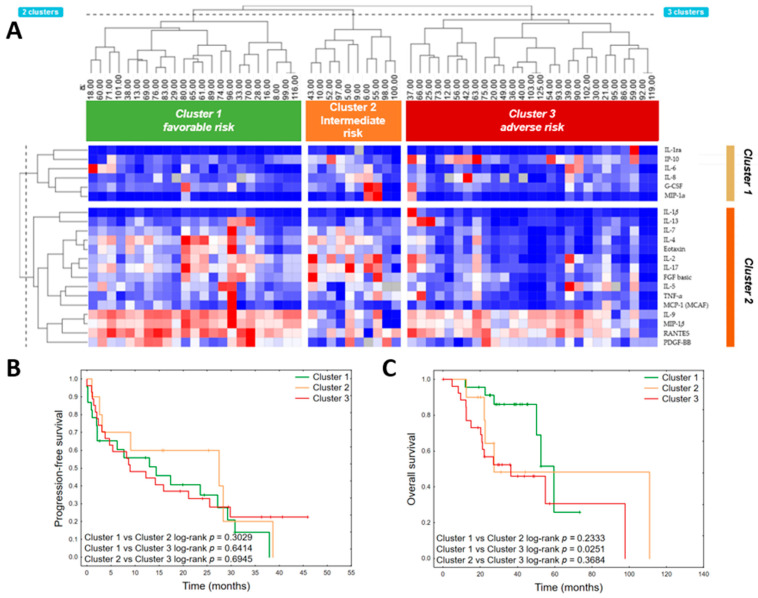
Heatmap with hierarchical clustering performed on cytokine values (**A**) to identify multiple myeloma patient groups. Expression levels of individual cytokines are represented by shades of blue to red in the central heatmap, with the highest values in dark red and the lowest in dark blue. Kaplan–Meier plots of progression-free survival (**B**) and overall survival (**C**) each identified cluster of patients.

**Table 1 jcm-11-00112-t001:** The characteristics of the study group, comprising 61 patients receiving bortezomib-based therapy. Unless otherwise specified, all data are presented as frequency and percentage (%).

Characteristics	Total
Number of patients	61 (100)
Gender	M: 32 (52.5)
	F: 29 (47.5)
Age at diagnosis	61.9 ± 11.3
mean + SD (range)	(38.3–83.7)
Bortezomib regimen:	
VCD	49 (80.3)
VMP	5 (8.2)
VTD	4 (6.6)
Other	3 (4.9)
Paraprotein	
IgG	33 (54)
IgA	14 (23)
LCD	14 (23)
Bone disease at diagnosis	36 (59)
Calcium > 2.75 mmol/L at diagnosis	12 (19.7)
HB < 10 g/dL at diagnosis	21 (34.4)
Creatinine > 2 mg/dL at diagnosis	10 (16.4)
International Staging System (ISS)	I-17 (27.9)
	II-13 (21.3)
	III-29 (47.5)
Beta2-microglobulin increased (>3mg/L)	40 (65.6)
LDH > 240U/L	7 (11.5)
Response to induction therapy	
CR	24 (39.3)
VGPR	13 (21.3)
PR	14 (23.0)
SD	7 (11.5)
PD	3 (4.9)
ASCT	23 (37.3)
Cytogenetics *	*N* = 33
t(11;14)	1 (3)
t(4;14)	5 (15.2)
t(14;16)	0
t(14;20)	0
del(17p)	3 (9.1)
amp(1q)	18 (54.5)
del(13q)	8 (24.2)

* Cytogenetics data were available for 33 patients (54.1%). In cytogenetic tests, at least 20 metaphases were analyzed; aberrations were positive if they were found in at least three metaphases. Abbreviations: ASCT—autologous stem cell transplantation; CR—complete response; LCD—light chain disease; PD—progressive disease; PR—partial response; SD—stable disease; VCD—bortezomib, cyclophosphamide, and dexamethasone; VD—bortezomib and dexamethasone; VGPR—very good partial response; VMP—bortezomib, melphalan, and prednisone; VTD—bortezomib, thalidomide, and dexamethasone.

**Table 2 jcm-11-00112-t002:** Univariate Cox regression analyses of basic clinical variables for progression-free survival and overall survival.

	PFS					OS				
Variable	Coefficient	*p*	HR	95% CI		Coefficient	*p*	HR	95% CI	
				Lower	Upper				Lower	Upper
ISS III	−0.13	0.680	0.88	0.48	1.60	0.78	0.060	2.18	0.97	4.90
ASCT	−1.11	0.001	0.33	0.17	0.64	−1.46	0.004	0.23	0.09	0.62
HB < 10 g/dL at diagnosis	0.12	0.720	1.13	0.59	2.17	0.12	0.778	1.13	0.49	2.58
Calcium > 2.75 mmol/l at diagnosis	0.46	0.213	1.58	0.77	3.23	−0.40	0.464	0.67	0.23	1.95
Creatinine > 2 mg/dL at diagnosis	−0.48	0.323	0.62	0.24	1.60	0.07	0.893	1.08	0.36	3.18
Bone disease	0.64	0.054	1.89	0.99	3.61	0.61	0.156	1.85	0.79	4.30
Age > 70	0.34	0.323	1.41	0.71	2.80	0.63	0.118	1.87	0.85	4.10

**Table 3 jcm-11-00112-t003:** Univariate Cox regression analyses of cytokine levels for progression-free survival and overall survival. Cytokine concentrations (pg/mL) were divided by 10 to allow a more straightforward interpretation of coefficient values.

	PFS					OS				
Cytokine	Coefficient	*p*	HR	95% CI		Coefficient	*p*	HR	95% CI	
				Lower	Upper				Lower	Upper
Eotaxin	0.0022	0.9558	1.0022	0.9279	1.0824	−0.0954	0.0620	0.9090	0.8224	1.0048
FGF basic	−0.0825	0.6462	0.9208	0.6475	1.3096	−0.2964	0.2059	0.7435	0.4697	1.1769
G-CSF	−0.0009	0.8658	0.9991	0.9882	1.0101	0.0002	0.9755	1.0002	0.9852	1.0155
GM-CSF	−1.7105	0.0694	0.1808	0.0285	1.1456	−4.9713	0.0643	0.0069	0.00004	1.3441
IFN-*γ*	−0.0230	0.8837	0.9773	0.7184	1.3296	−0.1371	0.5175	0.8719	0.5756	1.3206
IL-10	−0.0005	0.9984	0.9995	0.6270	1.5934	0.0049	0.0531	1.0049	0.9999	1.0099
IL−12 (p70)	0.6213	0.3417	1.8613	0.5172	6.6991	0.3990	0.7549	1.4903	0.1217	18.2445
IL-13	−1.9675	0.0185	0.1398	0.0272	0.7189	−4.4187	0.0087	0.0121	0.0004	0.3271
IL-15	0.0138	0.5837	1.0139	0.9652	1.0650	−0.0110	0.7869	0.9890	0.9131	1.0713
IL-17	−0.0975	0.5954	0.9071	0.6329	1.3001	−0.4021	0.1243	0.6689	0.4006	1.1170
IL-1ra	0.0012	0.8111	1.0012	0.9917	1.0107	0.0514	0.0029	1.0527	1.0177	1.0889
IL-1*β*	−1.6824	0.1799	0.1859	0.0159	2.1737	−1.3066	0.5299	0.2707	0.0046	15.9668
IL-2	−0.3222	0.6034	0.7246	0.2149	2.4429	−1.2680	0.1562	0.2814	0.0488	1.6234
IL-4	−0.1722	0.7859	0.8418	0.2430	2.9165	−1.6615	0.0364	0.1899	0.0401	0.9000
IL-5	0.1369	0.2194	1.1467	0.9217	1.4268	−0.0654	0.6848	0.9367	0.6829	1.2847
IL-6	0.0328	0.9321	1.0334	0.4860	2.1970	−0.2548	0.6893	0.7751	0.2223	2.7027
IL-7	0.0450	0.7981	1.0460	0.7411	1.4762	−0.4600	0.0497	0.6313	0.3988	0.9993
IL-8	0.1698	0.3739	1.1850	0.8151	1.7229	0.0972	0.7127	1.1021	0.6570	1.8487
IL-9	0.0217	0.1296	1.0219	0.9937	1.0509	−0.0149	0.3157	0.9852	0.9570	1.0143
IP-10	−0.0005	0.9984	0.9995	0.6270	1.5934	0.0049	0.0531	1.0049	0.9999	1.0099
MCP-1	0.0856	0.4049	1.0894	0.8906	1.3326	−0.2965	0.0992	0.7435	0.5226	1.0575
MIP-1*α*	−0.1895	0.3722	0.8273	0.5457	1.2545	0.0639	0.7731	1.0660	0.6903	1.6461
MIP-1*β*	0.1054	0.0990	1.1112	0.9804	1.2595	−0.0463	0.4618	0.9548	0.8441	1.0800
PDGF-BB	−0.0012	0.2900	0.9988	0.9965	1.0011	−0.0037	0.0316	0.9963	0.9930	0.9997
RANTES	−0.0007	0.1152	0.9993	0.9984	1.0002	−0.0010	0.0572	0.9990	0.9981	1.0000
TNF-*α*	−0.1612	0.3417	0.8511	0.6105	1.1866	−0.3491	0.1528	0.7053	0.4371	1.1382
VEGF	0.0022	0.7425	1.0022	0.9890	1.0157	−0.0395	0.1177	0.9613	0.9149	1.0100

**Table 4 jcm-11-00112-t004:** Final multivariate Cox regression analyses for PFS and OS of MM patients. Cytokine concentrations (pq/mL) were divided by 10 to allow a more straightforward interpretation of coefficient values.

Variable	Coefficient	*p*	HR	95% CI	
				Lower	Upper
PFS					
IL-13	−1.958	0.0302	0.1411	0.0240	0.8291
ASCT	−0.494	0.0065	0.3722	0.1826	0.7585
OS					
IL-1Ra	0.017	0.0091	1.017	1.004	1.030
IL-4	−1.828	0.0147	0.161	0.037	0.698
ASCT	−0.975	0.0007	0.142	0.046	0.438
bone disease	0.671	0.0059	3.826	1.471	9.949

## Data Availability

The data presented in this study are available from the corresponding author for request.

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
