# Peer review of "Pretreatment Serum Levels of IL-1 Receptor Antagonist and IL-4 Are Predictors of Overall Survival in Multiple Myeloma Patients Treated with Bortezomib"

_jcm, 2021, doi:10.3390/jcm11010112_

Round 1

Reviewer 1 Report

Mikulski et al. measured pre-treatment serum levels of cytokines in the serum of MM patient serum. The aim of the present study was to assess the impact of pretreatment serum levels of 27 selected cytokines on 
progression-free survival (PFS) and overall survival (OS) in MM patients before first-line therapy with bortezomib-based regimens. Serum cytokine levels were assayed with a Bio-Rad Bio-Plex Pro Human Cytokine 27-Plex Assay on the MAGPIX Multiplex Reader and the Bio-Plex® 200 System including IL-1β, IL-1Ra, IL-2, IL-4, IL-5, IL-6, IL-7, IL-8, IL-9, IL-10, IL-12, IL-13, IL-15, IL- 30, IL-17, Eotaxin, FGF, G-CSF, GM-CSF, IFN-γ, IP-10, MCP-1, MIP-1α, MIP-1β, PDGF-BB, RANTES, TNF-α, and VEGF. A total of 61 MM patients were examined. The authors report that three clusters of MM patients were identified with different cytokine profiles. In conclusion, serum pretreatment levels of IL-13 and IL-4 are predictors of better PFS and OS, respectively, whereas IL-1Ra pretreatment levels negatively impact OS in MM patients treated with bortezomib-based chemotherapy 

Line 94- Briefly is misspelled.

Section 3.4 - More details are needed to indicate how the MM patient population was stratified into three clusters. In addition, which cytokines were elevated in all patients and which were elevated in each of the three clusters.

Patient clinical demographics could be included in a Table.

Author Response

Reviewer 1.

R: Line 94- Briefly is misspelled.

Response: Corrected accordingly

R: Section 3.4 - More details are needed to indicate how the MM patient population was stratified into three clusters. In addition, which cytokines were elevated in all patients and which were elevated in each of the three clusters.

Response: Precisely, hierarchical clustering was made using one minus Pearson’s correlation matrix of included cytokines (as a measure of distance) with a complete linkage method. Generally, the third cluster presented the most distinctive cytokine profile pattern with decreased concentrations of proinflammatory cytokines (IL-2, IL-7, IL-9, IL-17, FGF basic, PDGF-BB) and chemokines (RANTES, MIP-1β, and Eotaxin). Furthermore, the third cluster had the lowest mean concentration of regulatory cytokine IL-4 and the highest mean level of IP-10. The first and second clusters had substantially similar cytokine patterns, although patients within the second cluster presented with the lowest concentrations of IL-9, MIP-1β, and RANTES, and the highest levels of FGF basic and MIP-1α.

As stated in the manuscript, detailed comparisons of concentrations of all cytokines between identified clusters of patients are included in Table S1. Appropriate explanations and descriptions were added to 3.4. section and Statistical analysis section. 

Table S1. Cytokine profile of three identified clusters of multiple myeloma patients. Data are presented as mean values and SD or median and interquartile range (IQR), depending on the variable distribution. P-values from the global test (ANOVA/Kruskal-Wallis test) are reported with post hoc comparisons (Tukey’s test/Duun’s test) if the p-value of the global test is significant.

Cytokine

Cluster 1

Cluster 2

Cluster 3

ANOVA p

Cluster 1 vs 2 p

Cluster 1 vs 3 p

Cluster 2 vs 3 p

Mean

SD

Mean

SD

Mean

SD

IL-13

2.97

2.06

2.44

0.93

2.40

2.43

0.6127

IL-17

23.64

5.76

26.04

10.06

16.44

6.54

0.0002

0.6367

0.0017

0.0014

PDGF-BB

3830.97

1305.15

2790.87

1697.88

2774.65

1064.63

0.0116

0.1019

0.0132

0.9995

Cytokine

Cluster 1

Cluster 2

Cluster 3

Kruskal-Wallis test p

Cluster 1 vs 2 p

Cluster 1 vs 3 p

Cluster 2 vs 3 p

Median

IQR

Median

IQR

Median

IQR

IL-1β

0.92

0.70

1.28

0.95

0.77

1.18

0.92

0.49

1.54

0.9509

IL-1ra

86.81

64.82

114.14

94.71

86.82

213.80

107.25

86.90

219.94

0.0717

IL-2

6.13

5.47

7.61

7.31

5.29

9.29

4.53

3.03

5.08

0.0008

1.0000

0.0068

0.0045

IL-4

7.42

5.85

9.44

7.81

5.65

8.52

3.41

2.74

5.54

0.0000

1.0000

0.0000

0.0024

IL-5

21.91

16.81

25.85

28.46

16.12

29.41

18.13

12.21

29.93

0.4486

IL-6

2.81

1.25

4.30

5.57

2.92

6.47

2.96

1.61

4.75

0.1287

IL-7

23.01

19.55

27.82

25.84

20.54

30.93

14.06

11.28

23.04

0.0004

1.0000

0.0026

0.0054

IL-8

8.56

6.45

10.60

11.83

9.05

23.20

10.29

5.43

16.79

0.2327

IL-9

519.49

476.02

570.64

302.43

123.32

462.24

457.36

372.13

490.54

0.0001

0.0002

0.0023

0.3957

IL-10

10.87

9.36

12.08

10.71

7.87

11.78

8.84

7.86

12.04

0.4269

IL-12 (p70)

1.26

1.26

1.61

-

-

2.65

2.65

4.50

1.000

IL-15

50.04

40.32

58.80

71.63

57.22

93.77

60.92

40.41

93.83

0.1763

Eotaxin

98.84

80.49

139.02

103.01

92.03

122.48

49.66

39.11

75.40

0.0000

1.0000

0.0001

0.0081

FGF basic

39.51

35.85

40.98

42.30

38.80

47.32

32.36

28.09

39.59

0.0007

0.5336

0.0194

0.0016

G-CSF

385.69

298.48

479.70

451.49

197.29

677.90

326.01

268.04

523.34

0.6091

GM-CSF

2.60

1.98

3.93

2.15

2.01

2.31

3.24

2.41

4.26

0.1279

IFN-γ

3.41

2.67

4.30

7.94

7.35

8.36

4.66

2.67

8.27

0.0564

IP-10

841.10

570.90

1018.39

1231.90

635.18

1513.88

1373.75

969.08

2395.38

0.0034

0.3437

0.0023

1.0000

MCP-1 (MCAF)

27.46

21.05

40.55

24.00

18.38

28.14

20.01

17.04

30.83

0.1231

MIP-1α

1.98

1.33

2.73

2.88

2.265

7.43

3.25

1.86

4.79

0.0058

0.0135

0.0357

0.9831

MIP-1β

140.95

127.75

150.37

92.10

63.20

118.96

125.82

109.31

135.94

0.0001

0.0001

0.0121

0.1388

RANTES

12785.10

10461.28

14692.82

9726.17

6770.13

11516.71

10534.02

8542.05

11858.24

0.0020

0.0079

0.0109

1.0000

TNF-α

23.21

21.67

30.83

31.24

23.90

35.41

21.37

17.23

28.94

0.0292

0.5292

0.3665

0.0307

VEGF

101.29

74.28

263.55

147.09

103.48

157.42

103.08

78.39

207.15

0.6090

R: Patient clinical demographics could be included in a Table.

Response: We suppose that this comment is also related to the cluster analysis of MM patients- such table is already presented as Table S2.

Table S2. Comparison of clinical variables between identified three clusters of multiple myeloma patients.

Variable

Cluster 1

Cluster 2

Cluster 3

p

N

%

N

%

N

%

ISS 3

7

30.4

8

80.0

13

52.0

0.0287*

HB < 10 g/dL at diagnosis

8

36.4

4

50.0

8

29.6

0.5627

Calcium > 2.75 mmol/l at diagnosis

4

18.2

1

12.5

7

25.9

0.6548

Creatinine > 2 mg/dL at diagnosis

2

9.1

2

25.0

6

22.2

0.4060

Bone disease

11

50.0

6

66.7

18

66.7

0.4527

Age >70

6

26.1

5

50.0

6

22.2

0.2386

Response to treatment: CR

7

30.4

5

50.0

11

40.7

0.5355

Response to treatment: ≥VGPR

13

56.5

6

60.0

17

63.0

0.8982

*Cluster 1 vs 2 p = 0. 008

Reviewer 2 Report

In their manuscript, Mikulski, et al. investigated the impacts of pretreatment serum levels of 27 cytokines on progression-free survival (PFS) and overall survival (OS) in multiple myeloma patients.

They employed a commercially available assay to detect 27 human cytokines and quantify their concentrations based on immune-assay results. This study included 61 patients and 80.3% of them received proteasome inhibitor bortezomib treatment. The authors performed univariate Cox regression analyses of pretreatment serum cytokine levels for PFS and OS. They identified that IL-13 serum level impacted both PFS and OS, whereas IL-1ra, IL-4, IL-7 and PDGF-BB significantly affects OS. Further analyses of these cytokines using Akaike information criterion (AIC) elimination procedure, the authors found that IL-13 cytokine level impacts PFS and IL-1Ra and IL-4 affects OS. Overall, this manuscript is well written and can be improved by modifying the following points.

Major points:

  1. In Table 1, the authors presented the overall information of patients enrolled in this study, including demographic, clinical, and laboratory characteristics. However, the authors either didn’t mention certain information showed in the table or only described in the text without showing in the table. For example, they listed beta2-microglobuline and LDH levels in those patients in Table 1; however, the authors didn’t describe them and why they are important in the text. The other way around, in the text, the authors indicated the response rate after primary treatment and the percentage of patients received ASCT; however, they didn’t include this information in the Table.
  2. In figure 3, the two-way hierarchical cluster analysis was shown to classify cytokine patterns. However, Figure 3A is very confusing, especially the title of each cluster. Please label the patient clusters and cytokine clusters accordingly in this figure.

Minor points:

  1. Please describe the abbreviations when they first showed in the text, such as VGPR in Line 142.
  2. Line 195-209 are redundant.

Author Response

Reviewer 2

R: Major points:

  1. In Table 1, the authors presented the overall information of patients enrolled in this study, including demographic, clinical, and laboratory characteristics. However, the authors either didn’t mention certain information showed in the table or only described in the text without showing in the table. For example, they listed beta2-microglobuline and LDH levels in those patients in Table 1; however, the authors didn’t describe them and why they are important in the text. The other way around, in the text, the authors indicated the response rate after primary treatment and the percentage of patients received ASCT; however, they didn’t include this information in the Table.

Response: The table has been completed as requested and lacking characteristics have been described in the text

  1. In figure 3, the two-way hierarchical cluster analysis was shown to classify cytokine patterns. However, Figure 3A is very confusing, especially the title of each cluster. Please label the patient clusters and cytokine clusters accordingly in this figure.

 Response: Patient clusters and cytokine clusters have been labeled accordingly in the Figure 3A.

R: Minor points:

  1. Please describe the abbreviations when they first showed in the text, such as VGPR in Line 142.

Response: All abbreviations are described in the revised version

  1. Line 195-209 are redundant.

Response: Corrected accordingly.

Reviewer 3 Report

This study is well written and well organized. The interesting point is the identification of the cytokine at the diagnosis prior to treatment with bortezomib-based regimen. I have only few suggestions.

1. Could you please identify the cut-off of the positive for high-risk cytogenetics test?

2. Some misspelling
- For the IL-1Ra, please spell consistently entire manuscript. (sometimes IL-1RA, IL-1ra, IL-1Ra)
- Line 48: plasma renal impairment  -> shoud it be renal impairment
- Line 281: SMM

3. As several studies indicated IL-1RA is an inibitor of IL-1 and IL-6 and it's known that IL-1 and IL-6 is an important cytokine of myeloma disease nad progression, could you explain why patients with higher level of IL-1RA led to worse OS?

4. Could you add the limitation of the study? How can we use these cytokines in the future perspective especially in the era of immunotherapy?

Author Response

Reviewer 3

  1. Could you please identify the cut-off of the positive for high-risk cytogenetics test?

Response: In cytogenetic test  at least 20 metaphases were analyzed and aberrations were considered positive if they were found in at least 3 metaphases.  We added this sentence in the table 1.

  1. Some misspelling
    - For the IL-1Ra, please spell consistently entire manuscript. (sometimes IL-1RA, IL-1ra, IL-1Ra)
    - Line 48: plasma renal impairment  -> shoud it be renal impairment
    - Line 281: SMM

Response: All misspellings were corrected.

  1. As several studies indicated IL-1RA is an inibitor of IL-1 and IL-6 and it’s known that IL-1 and IL-6 is an important cytokine of myeloma disease and progression, could you explain why patients with higher level of IL-1RA led to worse OS?

Response: IL-1Ra is an anti-inflammatory cytokine and potent inhibitor of IL-1 activity, as it binds to the receptor without activating it. In result, the prosperities of IL-1 could be reduced up to 95% [1]. IL-1Ra values in MM patients were reported to be elevated in the bone marrow [2]. In vitro studies showed that treatment of MM cells with IL-1Ra alone would inhibit IL-6 production, a central myeloma growth factor,  and myeloma cell growth, however it did not increased apoptosis of MM cells [1]. It was suggested that IL-1/IL-6 and IL-1Ra pathway is one of potential mechanism for the progression from MGUS to active MM [3]. Interleukin-1 levels increase as one proceeds from MGUS to SMM to active MM. In our study, in the third (adverse risk) cluster the concentration of IL-1Ra was the highest and the concentration of IL-6 was the lowest, but the differences were not statistically significant (p-value 0.19 and 0.48, respectively). Taken together, we hypothesize that  IL-1/IL-6 and IL-1Ra cascade probably is crucial in the process of conversion from MGUS to MM, whereas in active disease, other mechanisms, including acquisition of harmful genetic changes start taking over control. This finding is not clear  and needs further investigation.      

  1. Lust, J.A.; Lacy, M.Q.; Zeldenrust, S.R.; Dispenzieri, A.; Gertz, M.A.; Witzig, T.E.; Kumar, S.; Hayman, S.R.; Russell, S.J.; Buadi, F.K.; et al. Induction of a Chronic Disease State in Patients With Smoldering or Indolent Multiple Myeloma by Targeting Interleukin 1β-Induced Interleukin 6 Production and the Myeloma Proliferative Component. Mayo Clin. Proc.200984, 114–122..
  2. Cao, Y.; Luetkens, T.; Kobold, S.; Hildebrandt, Y.; Gordic, M.; Lajmi, N.; Meyer, S.; Bartels, K.; Zander, A.R.; Bokemeyer, C.; et al. The cytokine/chemokine pattern in the bone marrow environment of multiple myeloma patients.  Hematol.201038, 860–867.
  3. Xiong Y, Donovan KA, Kline MP, et al. Identification of two groups of smoldering multiple myeloma patients who are either high or low producers of interleukin-1. J Interferon Cytokine Res 2006; 26:83–95.

4. Could you add the limitation of the study? How can we use these cytokines in the future perspective especially in the era of immunotherapy?

Response: Our study has several limitations. It is not certain that the levels of IL-1Ra and IL-4 are genuinely predictive of response to bortezomib and not simply prognostic (i.e., indicative of a more refractory phenotype). To verify it, the response to the following lines of therapy should be evaluated, especially including other novel drugs. To do so, a much larger cohort of patients is needed with optimal stratification to available treatment options. Furthermore, equating baseline cytokine levels in MM patients with OS is a demanding task, and the results should be interpreted with caution. The cytokine levels should be further evaluated consecutively to identify changes related to the number of subsequent lines of therapy or acquiring resistance to particular drug classes. Even if they are only prognostic, they may help create more accurate prognostic models in the era of novel therapies (including immunotherapy) than our classical markers (e.g., R-ISS).
The study limitations are added in the end of the discussion.
Best regards 

Reviewer 4 Report

This study by Milkuski et al describes retrospective analysis of various baseline cytokine levels in patients treated with bortezomib inductions regimens. The data presented are clear, but I do have some concerns about the methods and conclusions. 

Specifically, the authors are attempting to correlate the baseline level of various cytokines and/or groups of cytokines with PFS and OS in newly diagnosed MM patients treated with a bortezomib based induction regimen. First, I would argue that in the era of novel therapies trying to equate baseline cytokine levels with overall survival is challenging to say the least and will require a much larger number of patients who are followed serially to ensure they are receiving adequate therapy (and normalized for lines of therapy, class exposure etc). Indeed only 38% of patients included in this analysis received ASCT (which is presumably does not represent all transplant eligible patients included in the study) and the relatively poor overall survival value of 50 months. This likely explains why significant changes in PFS do not correlate with OS in this dataset. Therefore I would focus on the PFS and omit the OS data and "predictive" modeling.

With regard to PFS this data is compelling, but to ensure that it is truly predicative of response to bortezomib and not simply prognostic (i.e. indicative of a more refractory phenotype) we would need to see if patients with elevations in these cytokines would be refractory to subsequent lines of therapy (especially lines that include other classes of novel therapies). I think including this data would be extremely powerful and would suggest predicative value in checking these cytokine levels. If they prove to be purely prognostic it still may inform a more accurate prognostic biomarker in our current era of therapy then our tradition markers (age, R-ISS stage, cytogenetics etc). 

Author Response

Reviewer 4

This study by Mikulski et al describes retrospective analysis of various baseline cytokine levels in patients treated with bortezomib inductions regimens. The data presented are clear, but I do have some concerns about the methods and conclusions. 

Specifically, the authors are attempting to correlate the baseline level of various cytokines and/or groups of cytokines with PFS and OS in newly diagnosed MM patients treated with a bortezomib based induction regimen. First, I would argue that in the era of novel therapies trying to equate baseline cytokine levels with overall survival is challenging to say the least and will require a much larger number of patients who are followed serially to ensure they are receiving adequate therapy (and normalized for lines of therapy, class exposure etc). Indeed only 38% of patients included in this analysis received ASCT (which is presumably does not represent all transplant eligible patients included in the study) and the relatively poor overall survival value of 50 months. This likely explains why significant changes in PFS do not correlate with OS in this dataset. Therefore I would focus on the PFS and omit the OS data and “predictive” modeling. With regard to PFS this data is compelling, but to ensure that it is truly predicative of response to bortezomib and not simply prognostic (i.e. indicative of a more refractory phenotype) we would need to see if patients with elevations in these cytokines would be refractory to subsequent lines of therapy (especially lines that include other classes of novel therapies). I think including this data would be extremely powerful and would suggest predicative value in checking these cytokine levels. If they prove to be purely prognostic it still may inform a more accurate prognostic biomarker in our current era of therapy then our tradition markers (age, R-ISS stage, cytogenetics etc). 

Response: We thank the Reviewer for the creative and clear review of our paper. As we agree with his/her comment we have concentrated on the PFS in our paper. However, we have not removed OS data, but we have added the proper comment in the discussion part of the revised paper. We agree with the reviewer that the analysis of response to the subsequent lines of therapy has essential value. A much larger cohort of patients is needed to conduct such a study with optimal stratification to available treatment options. In our dataset, 46 patients eventually progressed, and various treatment options were administered (daratumumab, KRD, clinical trials, etc.), complicating further analyses and reducing the statistical power of such results. Although an analysis of how these cytokines change their activity during different phases of treatment would be an exciting study to perform, it would likely require a different experimental design exceeding the scope of this analysis.

A comment was added to the discussion.

Reviewer 5 Report

The authors report on cytokine profiles in patients with newly diagnosed multiple myeloma undergoing bortezomib-based treatments. The manuscript is mainly well written and contains potentially new and relevant data. I have several comments.

The authors included ASCT in the analysis of different factors at diagnosis on PFS and OS. I rather doubt that the impact of ASCT might be analysed by the Kaplan-Meier method and the log-rank test from a statistical point of view in this setting. This issue should be discussed with a statistician.

The authors write that some patients were excluded due to pre-existing conditions (e.g. infections/allergy, page 2). Do the authors have information on the number of patients who were excluded because of these conditions?

It is stated that continuous variables are presented as mean ± standard deviation (page 3). Are all these variables normally distributed?

Page 4. Bone disease was identified only in 59% of patients. Did all patients undergo CT and/or MRI?

Page 4. ISS III and creatinine >2 mg/dl were associated with improved PFS but reduced OS. Do the authors have an explanation for these finding?

Page 6. This statement is not fully clear for me: `cytokine concentrations (pg/ml) were divided by 10 to allow a more straightforward interpretation of coefficient values´.

The authors found that `IL-13 is an independent predictor of longer PFS in MM patients treated with bortezomib-based chemotherapy´ (page 9). How do the authors explain the discrepancy between these finding and the data of Di Lulo showing a role for IL-13 in MM progression?

Have the authors analysed the associated between different cytokine levels and patient and disease characteristics such as isotype and genetics?

Some parts (e.g. introduction and discussion) should be shortened.

There are minor linguistic errors (e.g. Page 2, line 94)

Author Response

Reviewer 5

The authors report on cytokine profiles in patients with newly diagnosed multiple myeloma undergoing bortezomib-based treatments. The manuscript is mainly well written and contains potentially new and relevant data. I have several comments.

The authors included ASCT in the analysis of different factors at diagnosis on PFS and OS. I rather doubt that the impact of ASCT might be analysed by the Kaplan-Meier method and the log-rank test from a statistical point of view in this setting. This issue should be discussed with a statistician.

Response: Among statistical analyses that could be implemented in such analyses should be mentioned both competing risk models and KM method. Competing risks arise in clinical research when there are more than one possible outcome during follow up for survival data, and the occurrence of an outcome of interest can be precluded by another [1]. In this particular example, myeloma-related mortality may be of primary interest, but other causes of death can prevent its occurrence and deaths caused by reasons other than myeloma are typical examples of competing risks. To build model of cumulative incidence for event of interest, detailed data of causes of death are necessary to appropriate discrimination between myeloma/other causes of death. We did not have such data in our sources, we could obtain only the date of death.

Regarding our PFS analyses, it is essential to note that eventually, a progression of the disease occurred in 46/61 patients (75.4%- complete observations). Among competing events that should be recognized in the rest of the patients (censored observations) is death not related to myeloma, but there were only three cases of deaths without documented myeloma progression.

Taking this into account, we conclude that Kaplan-Meier estimates will not overestimate the incidence of myeloma progression significantly, and this kind of bias is neglectable. Regarding OS analysis we could not be able to gather information about the precise cause of death (myeloma-related/unrelated). In this case, statistical approaches such as Kaplan-Meier survival analysis and Cox proportional hazards regression seem appropriate.

Furthermore, the results of impact of ASCT on PFS/OS should be interpret with caution. Of course, other factors influence the effect of ASCT- i.e., only younger patients (<70 years old) without any contraindications to such procedure (other serious diseases) are qualified to this procedure, whereas both of these are considered as established prognostic factors of survival of myeloma patients.

  1. Zhang Z. Survival analysis in the presence of competing risks. Ann Transl Med 2017;5(3):47. doi: 10.21037/atm.2016.08.62

The authors write that some patients were excluded due to pre-existing conditions (e.g. infections/allergy, page 2). Do the authors have information on the number of patients who were excluded because of these conditions?

Response: We excluded two patients. The first one (ID 22) was excluded due to rheumatoid arthritis, the second one (ID 58) was excluded due to atopic dermatitis. The number of the excluded patients is added in the revised version.

It is stated that continuous variables are presented as mean ± standard deviation (page 3). Are all these variables normally distributed?

Response: This sentence was referred to the main body of the manuscript, and we provide in Table 1 only age at diagnosis as mean ± SD. Among cytokines, only IL-13, IL-17 and PDGF-BB were normally distributed. Description of statistical methods and Supplementary table 1 was updated.

Page 4. Bone disease was identified only in 59% of patients. Did all patients undergo CT and/or MRI?

Response: Yes, all of the patients underwent whole body low dose CT to asses bone disease (added to the methods).

Page 4. ISS III and creatinine >2 mg/dl were associated with improved PFS but reduced OS. Do the authors have an explanation for these finding?

Response: Actually, ISS III and creatinine >2 mg/dl were not associated with improved PFS. Despite the protective odds ratios (0.88 and 0.62, respectively), these results were not statistically significant (p=0.68 and p=0.32, respectively).

Page 6. This statement is not fully clear for me: `cytokine concentrations (pg/ml) were divided by 10 to allow a more straightforward interpretation of coefficient values’.

Response: We have incorporated in our PFS/OS analyses the levels of cytokines as continuous variables. The hazard ratio expresses the change for a 1 unit increase for cytokine level. Sometimes, especially when the cytokine has a large value (e.g., IP-10 mean concentration- 1222.3 ± 710.6), the change in OR for 1 unit is minimal. Dividing the original value by 10 or 100 enables easier visualization of the coefficients by limiting the number of leading zeros without change in significance level.

Coefficient

P-value

HR

95%CI - lower

95%CI - upper

IP-10

0.000491

0.053079

1.000491

0.999993

1.000988

IP10/10

0.004907

0.053079

1.004920

0.999935

1.009929

The authors found that `IL-13 is an independent predictor of longer PFS in MM patients treated with bortezomib-based chemotherapy’ (page 9). How do the authors explain the discrepancy between these finding and the data of Di Lulo showing a role for IL-13 in MM progression?

Response: Di Lulo et al. generated and characterized short-term primary BM-MSC (bone marrow mesenchymal stromal cells) cell lines from 3 healthy donors. IL-13 induced adhesion molecule upregulation and IL-6 production that was inhibited in the presence of an anti-IL-13 Ab. When IL–13-treated BM-MSCs were co-cultured with MM cells, they found significantly increased MM cell growth compared with untreated BM-MSCs. Taken together, we hypothesize that  IL-13/IL-6 cascade probably is crucial in the process of conversion from MGUS to MM, whereas in active disease, other mechanisms, including the acquisition of harmful genetic changes, start taking over control. This finding is not clear and needs further investigation. Proper explanation is added in the Discussion.

Have the authors analysed the associated between different cytokine levels and patient and disease characteristics such as isotype and genetics?

Response: Although a detailed analysis of how these cytokines change in various MM groups according to the cytogenetics/molecular classifications would be an exciting study to perform, it would likely require a large cohort of patients and d different experimental approach far exceeding the scope of this survival-oriented analysis. We obtained cytogenetics data from only 33 patients (54.1%), but we did not provide any results in the manuscript due to the low statistical power of such analyses.

Some parts (e.g., introduction and discussion) should be shortened.

Response: Introduction and discussion have been shortened as requested.

There are minor linguistic errors (e.g. Page 2, line 94).

Response: Corrected accordingly.

Round 2

Reviewer 4 Report

Agree with revisions.